# Alpha Lipoic Acid and Monoisoamyl-DMSA Combined Treatment Ameliorates Copper-Induced Neurobehavioral Deficits, Oxidative Stress, and Inflammation

**DOI:** 10.3390/toxics10120718

**Published:** 2022-11-24

**Authors:** Jayant Patwa, Ashima Thakur, Swaran Jeet Singh Flora

**Affiliations:** Department of Pharmacology and Toxicology, National Institute of Pharmaceutical Education and Research (NIPER-R), Lucknow 226002, India

**Keywords:** alpha lipoic acid, antioxidant, chelation, copper, MiADMSA, oxidative stress, neurotoxicity, rats

## Abstract

Copper (Cu), being an essential trace metal, plays several roles in biological processes, though exposure to Cu can be potentially toxic to the brain and a few other soft organs. In the present study, we investigated the effects of the combined administration of monoisoamyl 2, 3-dimercaptosuccinic acid (MiADMSA), which is a new chelator, and alpha lipoic acid (ALA) and an antioxidant that is made naturally in the body and is also found in foods, against Cu-induced oxidative stress in rats. Rats were exposed to 20 mg/kg copper sulfate for 16 weeks once a day via the oral route. After 16 weeks of exposure, animals were divided into different sub-groups. Group I was divided into three subgroups: Group IA, control; Group IB, MiADMSA (75 mg/kg, oral); Group IC, ALA (75 mg/kg, oral), while Group II was divided into four subgroups: Group IIA, Cu pre-exposed; Group IIB, Cu+ MiADMSA; Group IIC, Cu+ ALA; Group IID, Cu+ ALA+ MiADMSA. Exposure to Cu led to significant neurobehavioral abnormalities; treatment with MiADMSA, and in particular MiADMSA + ALA, significantly ameliorated the neurobehavioral parameters and restored the memory deficits in rats. Oxidative stress variables (ROS, nitrite, TBARS, SOD, catalase) and inflammatory markers (TNF-α, and IL-1β), which were altered on Cu exposed rats, also responded favorably to ALA+ MiADMSA combined treatment. Thus, combined administration of MiADMSA and ALA might be a better treatment strategy than monotherapy with MiADMSA or ALA against Cu-induced neurotoxicity, particularly in reducing oxidative stress, neurobehavioral abnormalities, and inflammatory markers.

## 1. Introduction

Metals have a remarkable role in human cellular physiology [1]. Among them, Cu is one of the essential metals known for its redox activity. It serves as an inorganic co-factor for various cellular enzymes and is involved in the regulation of cellular physiology in eukaryotes. Ceruloplasmin, superoxide dismutase (SOD), catalase, lyase, and beta-hydroxylase are well-known examples of Cu-containing enzymes [2]. The transition property of Cu is involved in the maintenance of redox cycling. Excessive Cu may cause dyshomeostasis of the redox cycle, which imparts a deleterious effect on the cell. Therefore, excess Cu buildup in the brain is linked with neurodegenerative disorders, including Wilson’s disease (WD) and Alzheimer’s disease (AD) [3,4]. Copper mining workers are at great risk due to the inhalation of copper dust. Studies on Cu mining workers linked it with several disorders, including anxiety, depression, and lung disease. Researchers have suggested chelation, along with antioxidant therapy, could be beneficial to mining workers against chronic Cu intoxication.

WD is the prototype of Cu toxicity; although it is a genetic disorder, the symptoms arise due to excessive deposition of Cu in targeted organs such as the brain, liver, and lungs, etc. [5]. Oxidative stress is the key mechanism suggested for Cu toxicity [6]. It is reported that the redox nature of Cu generates free radicals in the presence of hydrogen peroxide by the Fenton reaction [7]. The excessive ROS generation cripples the antioxidant defense systems, which might induce lipid peroxidation of the cellular membrane. Moreover, ROS are highly reactive towards amino acids, proteins, and nucleic acids; consequently, excessive ROS load degrades their activity [8]. ROS has a pleiotropic effect and is thus considered a key player in inflammation generation [9]. It is well established that higher Cu levels in the brain induce neuroinflammation [10]. 

Chelation therapy is an effective strategy to reduce the Cu load on soft tissues, but some limitation has been associated with it, such as adverse effects and metal redistribution [11]. Recent developments in chelation therapy have recommended a combination therapy with a chelating agent instead of monotherapy [12]. DPA is the first drug of choice for the treatment of Cu toxicity; however, its numerous side effects limit its use in the clinical management of Cu-associated disorders [13]. Investigations are continuing to provide safer treatment against Cu toxicity. MiADMSA is a well-known arsenic chelator and may be a future drug of choice against chronic arsenic toxicity [14]. Interestingly, during the preclinical investigation, we have noticed that, besides removing arsenic from the blood and soft tissues, it also depletes essential Cu levels [15]. This advocates its possible use against Cu toxicity [16]. Therefore, we have explored its effect against Cu toxicity and, interestingly, we found that MiADMSA can reduce Cu load from the brain and liver.Moreover, MiADMSA reduced the oxidative stress in a rat model [17,18]. ALA is a naturally occurring organosulfur compound in humans and animals. It is a low molecular weight substance readily absorbed from the diet, and due to its lipophilic nature it can easily cross the blood–brain barrier [19]. MiADMSA has thiol molecules, which are responsible for Cu chelation. Our previously published study on the effect of MiADMSA on other essential elements suggests that MiADMSA did not alter the essential elements such as iron, zinc, magnesium, and calcium in blood; however, Cu level was decreased [15]. Therefore, it could be a huge advantage over other chelating agents as it does not interfering with other bio-metals.

ALA is an antioxidant that is made naturally and is also known for its anti-inflammatory activity [20]. Besides these biological activities, it has some metal-chelating properties due to the presence of thiol moiety. The neuroprotective effects of ALA have been extensively explored in cell line and animal models [21,22]. Pandey et al. suggests that ALA treatment significantly reduced the metal load from the rat’s brain and reduced the oxidative stress marker from the brain [21]. Smirnova et al. has shown that ALA is effective against Cu toxicity in the Huh7 (hepatocyte) cell line [23]. The protective effects of individual molecules have been explored previously, yet no one has evaluated the combined effect with a metal chelator upon Cu toxicity. 

Thus, the current study was designed to investigate the combined administration of MiADMSA and ALA against chronic Cu-induced neurotoxicity in SD rats. The combined effect of MiADMSA and ALA were evaluated in terms of indicators of oxidative stress variables including ROS, TBARS, nitrite, catalase, SOD, neurobehavioral parameters, and inflammatory markers using the ELISA technique.

## 2. Material & Methods 

### 2.1. Animals

Male Sprague Dawley (SD) rats weighing approximately 80–100 g were used in the present study. The experimental procedures were carried out in accordance with the rules approved by the Institutional Animal Ethics Committee (IAEC) of the National Institute of Pharmaceutical Education and Research, Raebareli (NIPER-R), Lucknow, Uttar Pradesh, India. The IAEC approval number is NIPER/RBL/IAEC/10/MARCH 2018 (approval date, 10 March 2018). Rats were purchased from the CSIR-Central Drug Research Institute, Lucknow, India. The rats were provided animal feed (purchased from altromin, ImSeelenkamp Lage, Germany) and water (Aquapure) (Lucknow, India) ad libitum. They were acclimatized at the departmental animal house at 22 ± 2 °C, humidity 55 ± 5%, with 12-h light and dark cycles for 1 week before and during experiments. 

### 2.2. Chemicals

Copper sulphate (C8027-500G) and Alpha-lipoic acid (62320-5G-F) were purchased from Sigma-Aldrich (St. Louis, MO, USA). MiADMSA was received as a gift sample from Cadila Pharma, Ahmedabad, India. All other chemicals such as Ethanol (Merk 1.00983.0511), dichlorofluorescin diacetate (DCFDA) (D6883-250MG), Acetylthiocholine (C3389-500U), o- Dianisidine (D3252-25G), Hematoxyline (S059-500ML), Eosin (230251-25G), Nicotinamide adenine dinucleotide (N4505-500MG), Hydrogen Peroxide (1.93407.0521), 5,5′ -Dithiobis-(2-Nitrobenzoic Acid) (D8130-5G), Thiobarbituric acid (T5500-100G), Sodium dodecyl sulphate (L3771-1KG), Acetic acid (1.00063.2500), 1,1,3,3-Tetramethoxypropane (108383-500ML), Nitro blue tetrazolium chloride (N6876-250MG), Ammonium molybdate (A1343-100G), Sodium Phosphate Dibasic (S2364-500G) and Sodium Phosphate Monobasic (S3139-500G), all of analytical grade, were obtained commercially. 

### 2.3. Chelation Studies 

#### 2.3.1. Fluorescence Spectroscopy 

The fluorescence spectroscopic technique was used to analyze the ability of ALA to chelate Cu. An Agilent Carry fluorescence spectrophotometer was used for the fluorescence titration with a quartz cuvette, at room temperature. The excitation wavelength of ALA was set at λex = 222 nm, and the scanning for emission wavelength started from 265–430 nm with a slit width of 10 nm at 600 V. Cu in different concentrations (0–50 µM) was added to the solution of ALA (100 µM) in a 3 mL cuvette, and the spectra were acquired and plotted between wavelength vs. intensity. The fluorescence intensity of the emission spectra significantly changed with the addition of increasing concentrations of Cu to the ALA solution; this specifies the complex formation. 

#### 2.3.2. IR Spectroscopy 

IR spectra were recorded in a Bruker’s Fourier Transforms Infrared Spectra (FT-IR) spectrometer in which Attenuated Total Reflectance (ATR) mode was linked with OPUS software. Background was measured before the scanning of ALA and ALA with Cu complex. All the spectra were obtained after spectral resolution (24 scans) at room temperature (298 K) and after the baseline correction. This instrument describes the conformational changes in the structure of ALA after its interaction with Cu.

#### 2.3.3. NMR Spectroscopy 

All the NMR spectra were obtained from a Jeol 500 NMR spectrometer (Japan). ^1^H (proton) and ^13^C (carbon) NMR spectra were recorded using DMSO-*d*_6_ as a solvent, having the residual peaks at 2.5 ppm (quintet) for ^1^H NMR and 39.5 ppm (septet) for carbon NMR spectra. The integration of all the peaks were abbreviated as singlet (s), doublet (d), triplet (t) and multiplet (m). ALA and CuCl_2_ in equimolar concentrations were prepared in 0.6 mL DMSO-*d*_6_, and the obtained mixture was transferred to an NMR tube (5 mm diameter) using a micropipette of 1000 μL. Furthermore, the obtained spectra were analyzed using aJeol Delta NMR Processor (version 6.0.0) and Mestrenova (mnova) software (version 6.1.0). 

#### 2.3.4. Mass Spectroscopy 

All the High-Resolution Mass Spectra (HRMS) images were obtained in positive ion mode using an LC/QTOF high-resolution mass spectrometer (Agilent, Santa Clara, CA, USA) with a dual electrospray ionization source, having parameters of capillary voltage, 3500 V; source temperature, 35 °C; and gas flow, 11 L/min. The preparation of the compound (ALA) and its complex (ALA with Cu) was performed in LC-MS grade methanol. The spectra were recorded on an Agilent Mass Hunter LCMS Acquisition Console and were analyzed using Agilent Mass Hunter Qualitative Analysis software (version 10.0). 

### 2.4. Experimental Design 

Fifty-four (54) rats were randomly divided into two groups, each consisting of twenty-seven animals. A detailed schematic experimental design and treatment schedule is shown in Figure 1. Group I received normal drinking water, whereas Group II received 20 mg/kg CuSO_4_once a day through the oral route for 16 weeks. Following 16 weeks of exposure, animals were further divided into subgroups. Each subgroup consisted of 6 animals.


Group IA (*n* = 6): Saline (Control)Group IB (*n* = 6): MiADMSA (75 mg/kg, once, daily p.o.)Group IC (*n* = 6): ALA (75 mg/kg, once daily, p.o.)Group IIA, Cu-pre-exposed (*n* = 6): Saline (Copper sulfate 20 mg/kg)Group IIB, Cu-pre-exposed (*n* = 6): MiADMSA (75 mg/kg once daily, p.o.)Group IIC, Cu-pre-exposed (*n* = 6): ALA (75 mg/kg, once daily, p.o.)Group IID, Cu-pre-exposed (*n* = 6): ALA + MiADMSA (both 75 mg/kg, once daily, p.o.)


The doses of ALA, Cu, and MiADMSA were chosen based on our earlier published reports [17,21,24]. Blood samples were collected from cardiac puncture, and rats were sacrificed under euthanasia. The rats’ brains were isolated and carefully washed with ice-cold buffer. The brain homogenates were prepared for the evaluation of different biochemical parameters, and remaining tissue samples were preserved at −80 °C.

### 2.5. Assessment of Neurobehavioral Functions 

#### 2.5.1. Open Field Activity 

The open field test was used to evaluate the locomotor activity of rats using optovarimax apparatus (box measuring 90 cm × 90 cm × 30 cm). Rats were placed in the center of the open field and movements were automatically recorded during a 10-min testing period where the total distance travelled and resting time was recorded. The apparatus was cleaned using 70% ethanol and allowed to dry after each trial [25].

#### 2.5.2. Rotarod Test 

The Rotarod test was used to evaluate the muscle strength, motor coordination and balance of rats [26]. The whole experiment was divided into two phases: training and the final test. In the training session, each rat was placed individually into the compartment to habituate the rats on the stationary rod for 3 min. The next day, rats were acclimatized on the rotating road at 5 rpm/minute. During the final test, rats were placed on a rotating rod at varying speeds of 4–40 rpm/minute, and latency to falling was automatically recorded by instrument. 

#### 2.5.3. Elevated plus Maze 

The anxiety-like behaviors of the rats were assessed using elevated plus-maze apparatus [27]. The test apparatus has two open arms (50.8 × 12 cm) and two closed arms (50.8 × 12 × 40.6 cm), with a common center platform (10 × 10 cm). A camera (HD Logitech C525) was fixed on the top of the apparatus. At the beginning of the experiment, each rat was placed in the center of the apparatus facing an open arm. Every rat was allowed to explore the apparatus for 5 min, and their behavior was recorded. The number of entries and time spent in each arm were recorded by the video-tracking ANY-maze software. After 5 min, the rat was removed from the maze and the apparatus was cleaned with 70% ethyl alcohol and allowed to dry after each experiment. 

#### 2.5.4. Forced Swim Test (FST)

The depression-like behaviors of the rats were evaluated using a forced swim test [28]. An individual rat was placed in cylindrical tank (50 cm height and 30 cm width) containing water up to 30 cm (temperature of 24 °C ± 2 °C). The immobility time of the rat was recorded using a stopwatch. Immobility was defined as the total absence of movement in the whole body, that is, when the animal stopped struggling and remained motionless, floating on water, or when the animal was only doing the necessary movements to keep its head above the water. At the end of the trial, the rat was immediately removed from the tank, dried off with a paper towel, and returned to its home cage. 

#### 2.5.5. Nobel Object Recognition Test 

The NORT test was used to evaluate the effect of Cu exposure on memory function, as described by Antunes et al. [29]. The test was performed in a top-opened plastic box (65 L × 65 B × 45 H cm), with a camera (HD Logitech C525) fixed on the top of the apparatus. Software with nose-point detection was used for accurate analysis for evaluating the time spent with objects. The experiment was divided into the habituation phase, familiarization phase, and recognition phase. In the habituation phase, rats were taken to the experiment room and individually placed in the apparatus without objects and allowed to explore for 5 min. Animals were familiarized with two identical wooden blocks overnight. For the final recognition test, each rat was again placed in the apparatus with one familiar and one novel object for 10 min. The time spent with the novel object was recorded. The apparatus and objects were cleaned using 70% ethanol to minimize olfactory cues. 

#### 2.5.6. Passive Avoidance Test 

We performed the passive avoidance task method to evaluate the effect of Cu exposure and the beneficial effects of chelation as well as antioxidant therapy on memory function. We followed the method described by Elrod et al. [30]. We used Columbus instruments Passive/Active Avoidance System model PACS-30. The PACS apparatus (Shuttle Box) incorporated a dark chamber (27 cm × 14.5 cm × 14 cm) and a bright one (27 cm × 14.5 cm × 14 cm), which were separated by a guillotine door. Electric shocks in the dark chamber were provided via grid floor (0.5 mA, intensity for 5 second) by a standard simulator. The whole experiment was carried out on three successive days. On day 1, animals learned to enter a dark chamber from the light chamber. On day 2, after exploration, animals were placed in the light chamber. The door was opened, and the animals were allowed to enter the dark compartment. Once in the dark compartment, the door was closed, and an electric shock (0.5 mA, intensity for 5 second) was delivered through the floor. To evaluate memory, after 3 days, the transfer latency to the darkroom was again recorded. A maximum stop time of 300 s was considered for the rats staying in the light compartment. No electric shock was used during the retrieval test. 

### 2.6. Estimation of Brain AChE Activity 

The Ellman method was used to evaluate the brain AChE activity [31]. Briefly, brain samples were homogenized in the phosphate buffer and centrifuged at 10,000 rpm for 10 min. The supernatant was collected and used for AChE activity assessment. The whole experiment was carried out in 96-well plates, and each reaction well contained 2.7 mM ATCI (40 μL), 5 mM DTNB (100 μL), tested brain homogenate, and standard enzyme (80 μL). The absorbance was recorded in kinetic mode at 405 nm for 20 min at 1-min intervals. The percentage of AChE activity was calculated using the following formula.
%AChE activity = (Sample Absorbance/Control Absorbance) × 100(1)

### 2.7. Evaluation of Brain Oxidative Stress Markers 

#### 2.7.1. Reactive Oxygen Species 

The brain ROS levels were measured using the fluorescence DCFD method [32]. The reaction mixture contained 5 μL of brain homogenate, 10 μL of DCFDA solution, and 985 μL of phosphate buffer (pH 7.4). This solution was incubated at 37 °C for 15 min in dark conditions, and the product was measured on a multimode plate reader at excitation and emission wavelengths of 485 and 529 nm, respectively. The brain ROS levels were expressed by fluorescence intensity (FU)/mg of protein. 

#### 2.7.2. Thiobarbituric Acid Reactive Substances 

Brain TBARS levels were measured according to the protocol described by Ohkawa et al. [33]. In brief, 100 μL of brain homogenate was added in the test tubes, which contained 100 μL sodium dodecyl sulfate (8.1%), 300 μL acetic acid (20%), 300 μL TBA (0.8%), and 1.2 mL of distilled water. This solution was incubated at 90 °C for 60 min. The test tubes were then removed from the water bath and allowed to cool under running tap water. All the samples were centrifuged at 10,000 rpm for 10 min. Finally, the absorbance was recorded at 532 nm. 1,1,3, 3-tetra methoxy propane (97%) was used to prepare the standard curve, and the results were expressed as micromoles TBARS/mg protein. 

#### 2.7.3. Nitrite Levels

The brain nitrite levels were estimated according to the method described by Giustarini et al. [34]. The whole experiment was carried out in the 96-well plates. Each well contained 100 μL of Griess reagent and 100 μL collected brain supernatant. The reaction was incubated in the dark condition at room temperature with gentle shaking. Consequently, the developed chromogen was detected at 540 nm. Sodium nitrite was used to prepare the standard curve. Final brain nitrite content was expressed as micromoles/mg protein.

#### 2.7.4. Superoxide Dismutase Activity 

Brain SOD activity was measured according to protocols described by Kakkar et al. [35]. Briefly, brain homogenate was prepared in tris Hcl buffer and centrifuged at 10,000 rpm for 10 min, after which supernatant was collected. The final mixture contained 0.3 mL of PMS, 0.3 mL of NBT, 1.2 mL of sodium pyrophosphate, 0.2 mL of supernatant, 0.8 mL of distilled water, and 0.2 mL of NADH. The control was prepared with 1.2 mL of sodium pyrophosphate, 0.3 mL of PMS, 0.3 mL of NBT, 1 mL of distilled water, and 0.2 mL of nicotinamide adenine dinucleotide (NADH). The final product was then incubated at 37 °C for 90 s, and the reaction was stopped by adding 1 mL acetic acid. The mixture was allowed to stand for 10 min. The chromogen intensity was measured at 560 nm using UV-visible spectroscopy [35]. 

#### 2.7.5. Catalase Activity 

The brain catalase activity was assessed according to the protocol described by Gloth et al. [36]. Briefly, 200 μL of brain homogenate was incubated in 1 mL substrate (65 µmol/mL of H_2_O_2_ in 60 mmol/L sodium-potassium phosphate buffer pH 7.4) at 37 °C for 1 min. The reaction was stopped by adding 1 mL of 32.4 mmol/L ammonium molybdate. The yellow complex was determined at 405 nm.

#### 2.7.6. Total Protein Measurement 

Total protein content in brain homogenate was estimated according to the protocol described by Lowry et al. [37]. Briefly, 5 μL brain homogenate was incubated with solution D (containing 2% sodium carbonate, 0.4% sodium hydroxide, 2% sodium tartrate, and 1% CuSO_4_) for 10 min at 37 °C. Subsequently,200 μL Folin–Ciocalteu reagent was added in the reaction mixture and incubated further for 30 min at 37 °C. The absorbance was taken at a 652 nm wavelength. Pure bovine serum albumin (1 mg/mL) was used for standard curve preparation. 

### 2.8. Measurement of Neuroinflammatory Markers (TNF-α, & IL-1β) by ELISA 

Brain 8-OHdG, TNF-α, and IL-1β expressions were investigated using the ELISA technique, following the manufacturer’s instructions. The levels of 8-OHdG, TNF-α, and IL-1β in the brain was expressed in pg/mg of protein. Total protein was estimated using Lowry methods. 

### 2.9. Determination of Brain Cu Levels 

The brain Cu level was determined as described earlier [38]. In brief, brain tissue samples were digested using the conventional acid digestion method. The Cu content in the digested samples was measured using inductively coupled plasma mass spectrometry (ICP-OES, Model Optical emission spectrometer-Optima 5300 V by PerkinElmer).

### 2.10. Statistical Analysis 

All results were expressed as the mean ± standard error of mean (SEM). Statistical differences amongst the groups were analyzed by one way ANOVA followed by multiple comparisons with Tukey’s test, using Graph Pad (Graph Pad Prism version 6.0). A *p*-value of *** *p* < 0.001, ** *p* < 0.01, and * *p* < 0.05 was considered as statistically significant when copper-exposed animals were compared with normal control. On the other hand, $$$ *p* < 0.001, $$ *p* < 0.01, and $ *p* < 0.05 were considered statistically significant when MiADMSA- and ALA-treated animals were compared to the copper control.

## 3. Results 

### 3.1. ALA/Cu Chelation Studies 

The Cu chelation activity of the ALA was investigated using different spectroscopic techniques, including fluorescence, NMR, IR, and mass spectroscopy. We recorded the fluorescence spectra of ALA (100 μM concentration) and the successive addition of varying concentrations of Cu (0 to 50 μM). We found the fluorescence intensity of ALA was increased with the successive addition of Cu (Figure 2). Moreover, the bathochromic or red shift was observed from 337 nm to 367 nm, along with the hyperchromic shift. IR spectroscopic studies were also performed to obtain information regarding the coordination chemistry of Cu ions with ALA. In the spectra, the absorption band at 1703 cm^−1^ was attributed to carbonyl (C=O) stretching vibration in the ALA, and this band was almost disappeared and shifted to 1693 cm^−1^ in the ALA/Cu complex. This phenomenal change in the peak indicates that the carbonyl group might be involved in metal ion ligand complex formation. The OH vibration peak appeared at 3337 cm^−1^ and was shifted to 3316 cm^−1^ upon the addition of Cu (Figure 3). Further, to explore the mechanism of the interaction of ALA with Cu, ^1^H, and ^13^C, NMR studies were performed in DMSO-*d*_6_. The^1^H NMR spectra showed a singlet at 11.95 for the –COOH group, while the aliphatic–CH_2_protons come in the range of 1.3 ppm to 3.6 ppm. After the addition of Cu, the peak at 11.95 ppm was shifted to 11.83 ppm (Figure 4). In the ^13^C NMR, there was a chemical shift at 174.879 ppm for the carbonyl group, which disappeared upon the addition of Cu (Figure 5). The product was also confirmed by its HRMS calculated for ALA (C_8_H_14_O_2_S_2_): 206.0435 (M^+^), found 206.0421 and for ALA with Cu (C_8_H_14_CuO_2_S_2_): 268.9731 (M^+^), found 268.9722 (Figure 6). An outcome of these analyses suggests the Carbonyl group might have participated in the formation of the ALA/Cu complex.

### 3.2. Effect of ALA& MiADMSA on Brain Cu, Serum Ceruloplasmin Levels, and AChE Levels 

Chelation studies using different spectroscopic methods confirmed that MiADMSA and ALA may bind to Cu. However, to determine their chelating potential in a biological system, we need to determine the amount of Cu eliminated by the chelator from the tissues. In line with this, we measured the Cu level in the brain. The metal estimation data revealed that Cu level was significantly increased in the Cu exposed in comparison to the control one. It is interesting to note that in MiADMSA-treated rat brains, Cu level was significantly declined, whereas ALA-treated rats showed no significant recoveries (Figure 7A). However, combined treatment of MiADMSA and ALA showed a more pronounced recovery. Further, the Cu toxicity was confirmed by the serum ceruloplasmin levels. Ceruloplasmin is considered a surrogate marker of Cu toxicity. We have noted that ceruloplasmin level was significantly increased in the Cu-treated rats compared to the control rats. Marginal improvement was observed in the MiADMSA and ALAindividually-treated rats. However, MiADMSA and ALA combined administration significantly restored the ceruloplasmin level (Figure 7B). Brain AChE activity was also significantly decreased in Cu-challenged rats. However, our intervention improved the brain AChE activity, but the improvement was statistically nonsignificant (Figure 7C).

### 3.3. Effect of ALA and MiADMSA on Locomotors Activity 

Obtained spontaneous locomotor activity results depicted in Figure 8 suggest that the distance travelled by rats were significantly decreased (Figure 8B), whereas the resting time (Figure 8C) was significantly increased in the Cu-treated group in comparison to normal controls (Figure 8). Further, this observation was confirmed by the tracking plots shown in Figure 8A. It is clearly evident from the figure that Cu-exposed rats showed significantly less movement compared to the controls. MiADMSA and ALA treatment significantly ameliorated the observed alterations in behavioral activity, with the combination of MiADMSA and ALA producing the most robust effect. 

### 3.4. Effect of ALA and MiADMSA on EPM Performance 

The EPM test is widely used to evaluate anxiety-like behavior; herein, we used this test to evaluate the effect of Cu exposure on rat’s behavior and the impact of our intervention. Figure 9 illustrates the results from the EPM test. Cu-exposed rats spent more time in the closed arms (Figure 9B) and exhibited a higher level of entries into the closed arms (Figure 9C). The indictors of entries and time suggest Cu-exposed rats were demonstrating anxiety compared to controls. MiADMSA and ALA treatment significantly improved the EPM performance, confirmed by increased open arm entries, with the combination of MiADMSA and ALA producing the greatest effects.

### 3.5. Effect of ALA and MiADMSA on Memory Functions and Other Neurobehavioral Activities 

PACS and NORT tests were used to evaluate the memory functions in Cu-exposed rats (Figure 10A,B). A passive avoidance test result suggests that chronic Cu exposure significantly increased the transfer latency to the dark chamber. This could be correlated with memory dysfunctions. Similarly, in the NORT test we observed that rats spent less time with the novel objects as compared to the control rats. Interestingly, our proposed treatment improved the memory indicating parameters, but the changes were not significant in the individual treatments groups. However, the combined treatment of MiADMSA and ALA restored the memory parameters significantly. 

FST was used to evaluate the depressive behavior of Cu-exposed rats and the protective effect of MiADMSA and ALA on depressive behavior (Figure 10C). We observed that immobility time was significantly higher in the Cu-exposed rats, which is a key indicator of depressive behavior. We observed that immobility was significantly ameliorated by our intervention. However, MiADMSA and ALA treatments induced a more significant protective effect as compared to the individual treatments. The rotarod test was carried out to evaluate the motor coordination function in the Cu-treated rats (Figure 10D). We found that Cu-exposed rats spend less time on the rotating rod while the run time was significantly higher in all the interventions groups. The outcomes of the behavioral analyses suggest that combination therapy could be more effective in terms of improving memory deficits and other neurobehavioral functions.

### 3.6. MiADMSA and ALA Therapy Mitigates the Oxidative Stress

Cu is a redox-active metal; in particular, its transition properties play a key role in the generation of ROS, which is required for maintaining the redox biology of the cell. However, an excess generation of ROS can be potentially toxic to the cell. ROS are highly reactive and can bind to the lipid molecules and cause consequential damages to the lipid membranes. We noticed that Cu exposure significantly increased the ROS and nitrite level in the brain compared to the control rats (Figure 11A,B). As a result, ROS generation causes TBARS formation in the brain, which is an end product of lipid peroxidation (Figure 11C). However, MiADMSA and ALA therapy decreased the ROS, nitrite, and TBARS levels in the brain in comparison to Cu-treated rats. Interestingly, MiADMSA and ALA combined treatment reduced all the raised ROS, nitrite, and TBARS levels more significantly. 

### 3.7. MiADMSA and ALA Restored the Altered Antioxidant Enzymes 

In response to the raised ROS level, cells are equipped with antioxidant enzymes that counter the toxic responses of ROS by neutralizing them. Therefore, we explored the level of the antioxidant enzymes such as SOD and catalase in the Cu-exposed rats (Figure 12A,B). Our outcomes indicate that Cu treatment significantly decreased the level of antioxidant enzymes such as SOD and catalase in the brain compared to the control rats. Treatment with MiADMSA and ALA significantly restored the SOD and catalase levels compared to the Cu control group. However, the best improvement was observed in the MiADMSA and ALA combined treated rats.

### 3.8. Effect of ALA and MiADMSA on Brain Inflammatory Markers 

Further, ROS generation was confirmed by the 8-OHdG level. The 8-OHdG is an end product of ROS generation; therefore we measured it in the brain samples using the ELISA technique. The outcomes are shown in Figure 13A. We noticed that Cu exposure significantly raised the 8-OHdG levels in the brain, and our interventions with ALA and MiADMSA reduced the 8-OHdG levels in the brain; however, more pronounced effects were observed in those who received combination therapy. ROS are key signaling molecules that play an important role in triggering the inflammatory cascade. As we observed, Cu exposure significantly increases the ROS level in the brain, so it may induce neuroinflammation. Hence, we investigated the TNF-α and IL-1β level in the brain, which are key neuro-inflammatory markers (Figure 13B,C). Our ELISA results suggest that TNF-α, and IL-1β levels were significantly increased in the Cu-treated rat brain compared to the control rats. MiADMSA and ALA significantly reduced the expression of both neuroinflammatory markers in the brain. When compared to monotherapy, it was found that combination therapy elicits more significant recovery in term of reducing the expression of these inflammatory markers. 

## 4. Discussion 

Patients with neurological complications show higher Cu levels, which indicates its implication in neurological disorders such as AD, PD, and WD. Cu-induced amyloid-beta aggregation is a well-known mechanism postulated for AD pathogenesis [39]. Clioquinol and PBT-2 are two Cu chelators that have shown promising recovery in AD pathogenesis in pre-clinical investigation. Even phytomolecules, such as curcumin and resveratrol, for the treatment of Cu (II) toxicity, show promising efficiency in transgenic animal models due to their metal-chelating as well as their antioxidant properties. However; MiADMSA is also gaining recognition as a potential arsenic chelator [40]. Monotherapy of MiADMSA and ALA against Cu toxicityhas already been explored, yet no one has evaluated the effect of MiADMSA and ALA combination therapy against Cu toxicity in rats. Therefore, the current investigation was aimed at elucidating the combined effect of MiADMSA and ALA in chronic Cu-intoxicated rats.

We have reported the Cu chelation activity of MiADMSA in a previous investigation, where we used fluorescence and NMR spectroscopic methods. Our investigation suggests that the thiol molecules in MiADMSA were able to chelate with Cu [18]. However, the ALA chelation activity with Cu is obscure. The current investigation was undertaken to evaluate the Cu chelation activity of ALA using different spectroscopic methods. We took the fluorescence spectra of ALA alone, along with ALA with Cu. It has been found that when we added the increasing concentration of Cu into the ALA, the fluorescence intensity of ALA was increased, which indicate that ALA is binding to the Cu. The possible explanation for the increase in the fluorescence intensity after adding Cu might be due to the inhibition of the photo-induced electron transfer (PET) process in the complex, resulting in the enhancements of fluorescence intensity [41]. Moreover, this finding was further validated using IR spectroscopy. We found the C=O peak of the acid group present in ALA at 1703 cm^-1^ in spectra, and interestingly, when we added Cu to the ALA, this peak almost vanished and shifted to 1693 cm^−1^, which could be due to the binding of Cu to ALA. The following changes were further confirmed by ^1^H and ^13^C NMR spectroscopy, which gives information regarding the changes in the micro-environment of ALA. The outcome of the NMR study suggests that upon the addition of Cu, the peaks in the proton NMR were shifted; however, in the carbon NMR the carbonyl carbon peak vanished. This phenomenon has frequently been observed once a ligand binds to metal due to the intrinsic heterogeneity, such as the local structural variations of their surfaces [42]. Our spectroscopic investigations strongly support that Cu is chelating to ALA. We further used mass spectrometry for final confirmation because if Cu is chelating to ALA it will form a Cu + ALA complex. Therefore, we recorded the mass spectra of ALA and ALA + Cu. We found a strong peak in spectra, which indicates ALA is forming a complex with Cu. These observations encouraged us to see the beneficial effects of MiADMSA and ALA in the animal models.

We observed that treatment using MiADMSA and ALA significantly reduced the elevated brain Cu levels, although a combined treatment of MiADMSA and ALA could reduce more significantly as compared to an individual one. It has been reported that both MiADMSA and ALA have thiol moiety, which might be responsible for the Cu chelation [43]. Our previous reports also suggested that combined treatment using MiADMSA and ALA is more effective in reducing the heavy metal loads from the soft tissues [44]. Increased Cu level in the brain and CSF of neurodegenerative disorder patients indicates its involvement in the development as well as the progression of neurodegenerative disorders [4,45]. Recent publications and epidemiological studies suggest that Cu could be a contributing risk factor for AD [46]. The various types of neuronal damages observed in AD, including lipid peroxidation, nucleic acid oxidation, and reduced antioxidant potential, are well established and are directly or indirectly associated with Cu-induced oxidative stress generation [47]. Elevated Cu levels in the brain are considered a risk factor for AD-associated memory impairment and are also linked with memory impairment. Interestingly, chronic Cu exposure causes significant memory impairment, which was confirmed by the passive avoidance test. The Cu-intoxicated rats showed decreased latency to enter the dark compartment as compared to the control rats. We have not observed significant improvement in the rats that received monotherapy of ALA and MiADMSA. However, the transfer time was significantly improved in the MiADMSA and ALA-treated animals. The possible explanations for this could be due to the combination approach in which MiADMSA potentially decreased the Cu load, and ALA is a potent antioxidant that rescued neuronal cells from oxidative damages. ALA has also been considered a promising neuroprotective agent; recent reports suggest that ALA improves the neurotransmitter levels in the brain and thus improves the memory function of the animal models [48].

We noticed that Cu exposure causes a significant alternation in the other neurobehavioral parameters, such as open field activity, EPM performance, FST, and rotarod test. Interestingly, all reductions in neurobehavioral function were reversed by each of the intervention groups. The following recoveries might be due to the decreased Cu load and oxidative stress from the brain. MiADMSA is a known chelating agent, while ALA is a potent antioxidant. There is no direct evidence of MiADMSA and ALA combined effect on Cu toxicity, but it can be correlated with our previously published reports in which we reported that MiADMSA and ALA combined treatment was effective against lead-induced neurodegeneration in rats [21]. These neurobehavioral changes could be correlated with the altered brain AChE level. It is reported that a normal AChE level is required for smooth neurobehavioral function, whereas an abnormal level of AChE level may impart neurobehavioral abnormalities [49]. Our AChE estimation result suggests that Cu exposure decreased the AChE level, while treatment with MiADMSA and ALA treatment restored the AChE level. Therefore, the restoration of normal AChE levels ameliorates the neurobehavioral changes associated with Cu exposure. Further, an abnormal neurobehavioral function can be correlated with oxidative stress. We observed that chronic Cu exposure increased reactive oxygen species levels (ROS/RNS) and concomitantly reduced the levels of the antioxidant enzymes (SOD, catalase) in the brain. Oxidative stress is a key underlying mechanism suggested for Cu toxicity [50]. It is reported that Cu exposure increased ROS load by the Fenton-like reaction in the presence of H_2_O_2_ [7]. Excessive ROS generation disturbed the antioxidant potential, which might be the reason for the decreased level of antioxidant enzymes in the brain [51]. Interestingly chelation and antioxidant therapy restored the altered oxidative stress variables by reducing the Cu and ROS load. The ROS scavenging property of ALA has been widely accepted [52]. ROS has pleiotropic effects in the cell and is considered a key initiator of the inflammation process in tissue [9]. Cu-exposed rats showed increased TNF-α and IL-1β levels in the brain, which are the key markers of neuroinflammation. The increased inflammatory responses could be correlated to the activation of the NF-κB pathway in the brain. Persichini et al. reported that Cu activates the NF-κB pathway in-vivo. The NF-κB pathway plays a key role in inflammation [53]. Brain inflammation was mitigated by chelation and antioxidant treatments. We noted that MiADMSA and ALA treatment significantly reduced the TNF-α and IL-1β levels. The protective effect of MiADMSA and ALA might be due to antioxidant potential and Cu chelation. MiADMSA expels the Cu from the cell and ALA reduced the cellular oxidative damage by decreasing ROS levels in the brain and boosting the antioxidant capacity. Our findings can be correlated with the earlier findings of Dwivedi et al., who suggested that ALA treatment significantly reduced ROS-mediated inflammatory responses in rat brains [44]. Cu chelation therapy might be helpful during the early diagnosis of neurological disorders. Chelation therapy might reduce the risk of worsening the disease. 

One of the limitations of this was that no dose-response approach of ALA and MiADMSA, respectively, was attempted, primarily for the reason that earlier studies from our group have reported that ALA administration provided a dose-dependent protective effect in terms of the amelioration of oxidative stress parameters, and we considered that 75 mg/kg is the ideal dose to be administered alone and in combination with MiADMSA. In the case of MiADMSA, we have noted that MiADMSA at a dose higher than 75 mg/kg (example 150 mg/kg) may lead to some liver related injuries, as well as a significant loss of other essential metals. Thus, based on our previous experience, we opted for a single dose of MiADMSA along with ALA as it has anti-oxidant, metal chelation, and anti-inflammatory activity. In future we plan to study the dose-dependent effects of combination therapy using variable doses of MiADMSA and/or ALA. From the data available, it can be assumed that the increased efficacy of the combination therapy in the present is principally due to the simple additive effects of the two compounds, and no evidence of synergistic effect of the compounds were seen. It can thus be suggested that 75 mg/kg ALA plus 75 mg/kg MiADMSA, in general, causes a more pronounced responses than each of them alone at a dose of 75 mg/kg.The results are thus expected findings, and as the present experiments did not include 150 mg/kg of ALA or MiADMSA, it is difficult to conclude that combined administration of MiADMSA and ALA might be a better treatment strategy than monotherapy.

## 5. Conclusions

In conclusion, the current investigation highlights the protective effects of MiADMSA and ALA post chronic Cu-induced neurotoxicity. We report that chronic Cu exposure led to a significant Cu deposition in the brain, resulting in increased oxidative stress, inflammation, and neurobehavioral abnormalities. The result of our study suggests that combined administration of MiADMSA and ALA elicited more significant recoveries compared to monotherapy with these compounds. Our study also recommends including antioxidant during chelation therapy against copper intoxication. Combination therapy would thus be beneficial for Cu-induced redox imbalance and inflammatory instigations.However, a more detailed mechanistic investigation still needs to be performed to further explain the exact molecular mechanism involved in these observations.

## Figures and Tables

**Figure 1 toxics-10-00718-f001:**
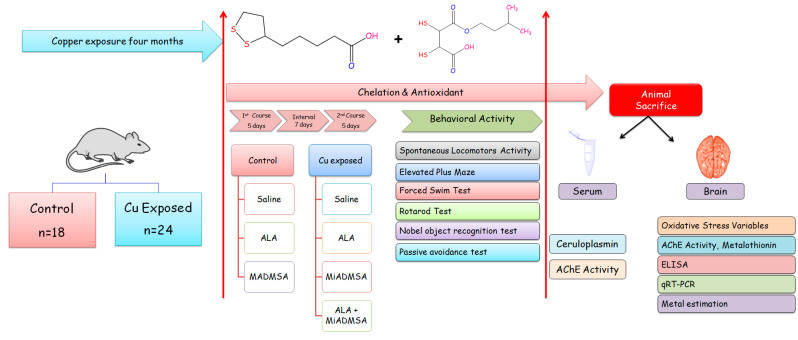
Graphical experimental overview where rats were randomly divided into two groups, each consisting of twenty-seven rats. Group I received normal drinking water, whereas Group II received 20 mg/kg CuSO_4_ once a day through the oral route for 16 weeks. Following 16 weeks of exposure, animals were further divided into subgroups. After the treatment, animals underwent neurobehavioral analysis. Thereafter, animals were sacrificed and blood and brain samples were collected for further biochemical estimation.

**Figure 2 toxics-10-00718-f002:**
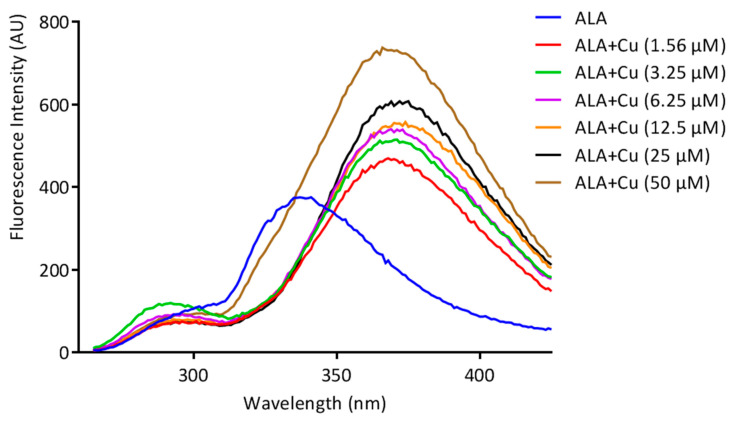
Fluorescence titration of ALA (100 µM) with increasing concentrations of Cu (0–50 μM). A bathochromic or red shift was observed from 337 nm to 367 nm, along with the hyperchromic shift.

**Figure 3 toxics-10-00718-f003:**
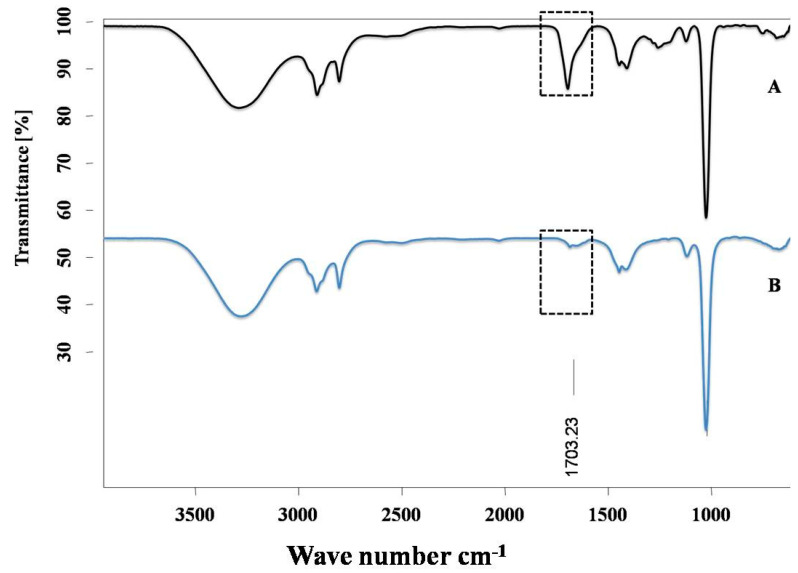
IR spectra of (**A**) ALA and (**B**) ALA with Cu. Carbonyl (C=O) peak present in ALA comes at 1703 cm^−1^ and almost disappears after the addition of Cu, suggesting that the carbonyl group might be responsible for metal ion ligand complex formation. The dotted box indicates changes in the peaks.

**Figure 4 toxics-10-00718-f004:**
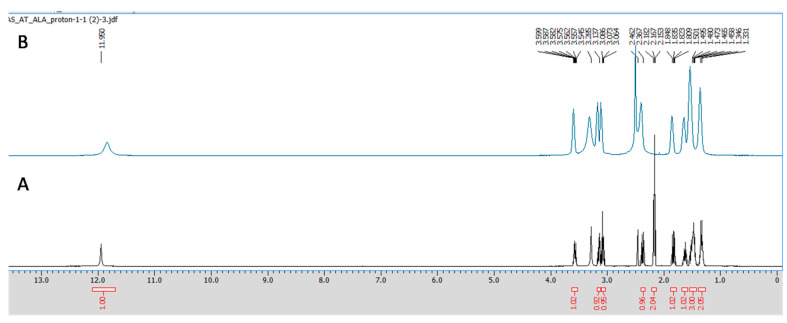
^1^H NMR of (**A**) ALA and (**B**) ALA with Cu after 5 min in DMSO-*d*_6_ at 500 MHz. A singlet peak at 11.95 was attributed to the -COOH group, and after the addition of Cu, the peak was upward shifted to 11.83 ppm.

**Figure 5 toxics-10-00718-f005:**
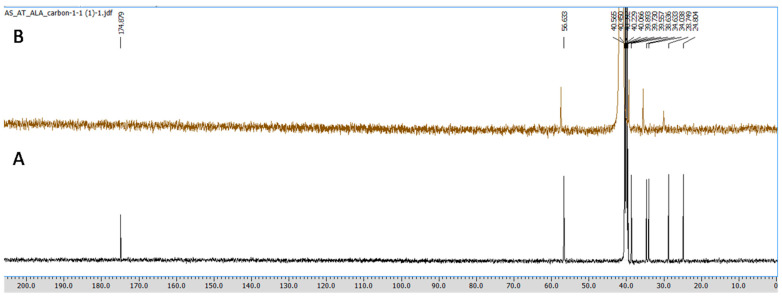
^13^C NMR of (**A**) ALA and (**B**) ALA with Cu after 5 min in DMSO-*d*_6_ at 125 MHz. The carbonyl peak comes at 174.87 ppm in the ^13^C NMR spectra of ALA, and after the addition of Cu, the peak vanished, indicating that the carbonyl group was involved in the metal chelation.

**Figure 6 toxics-10-00718-f006:**
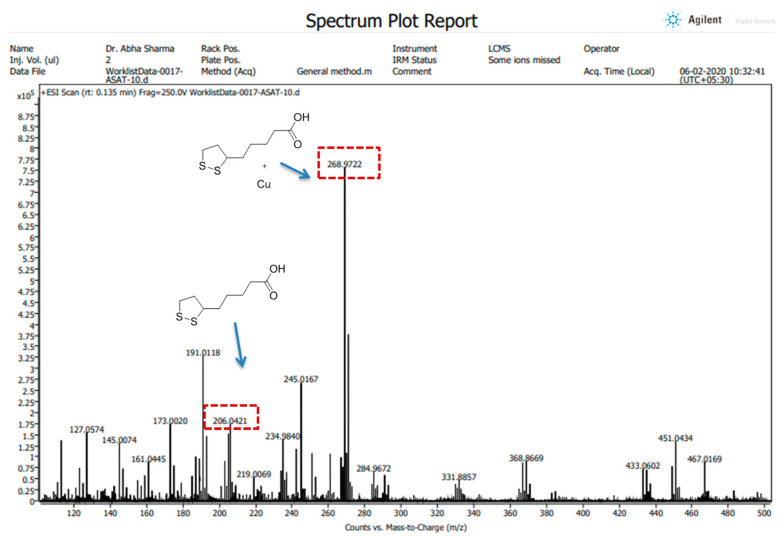
HRMS spectra of ALA and ALA with Cu. The complex between ALA with Cu was further confirmed by HRMS and shows the mass adduct formation of ALA alone at 206.04 and with Cu at 268.97.

**Figure 7 toxics-10-00718-f007:**
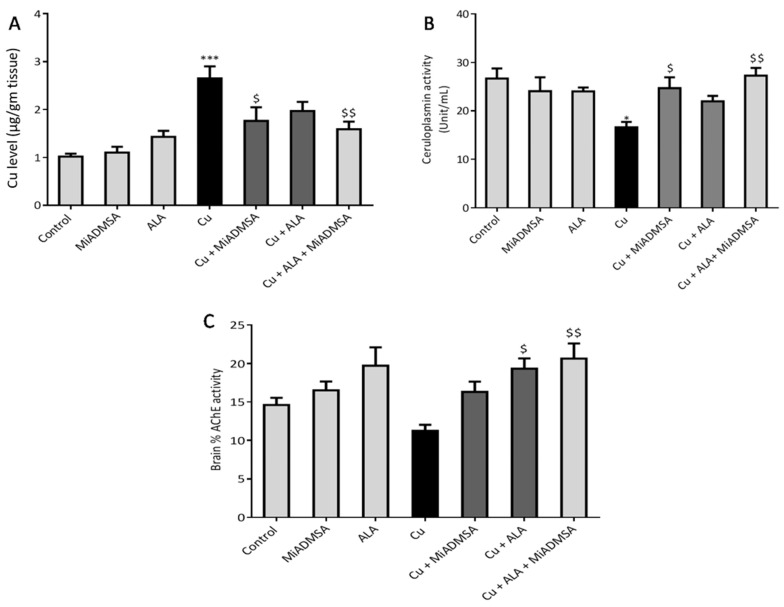
Effect of combination therapy of MiADMSA and ALA on ceruloplasmin level, brain Cu, and AChE levels. (**A**) Brain Cu level; (**B**) Serum ceruloplasmin level; (**C**) Brain AChE levels. All the values are expressed as mean ± S.E.M. (*n* = 6) * *p* < 0.05, *** *p* < 0.001 vs. control and $ *p* < 0.05, $$ *p* < 0.01 vs. Cu.

**Figure 8 toxics-10-00718-f008:**
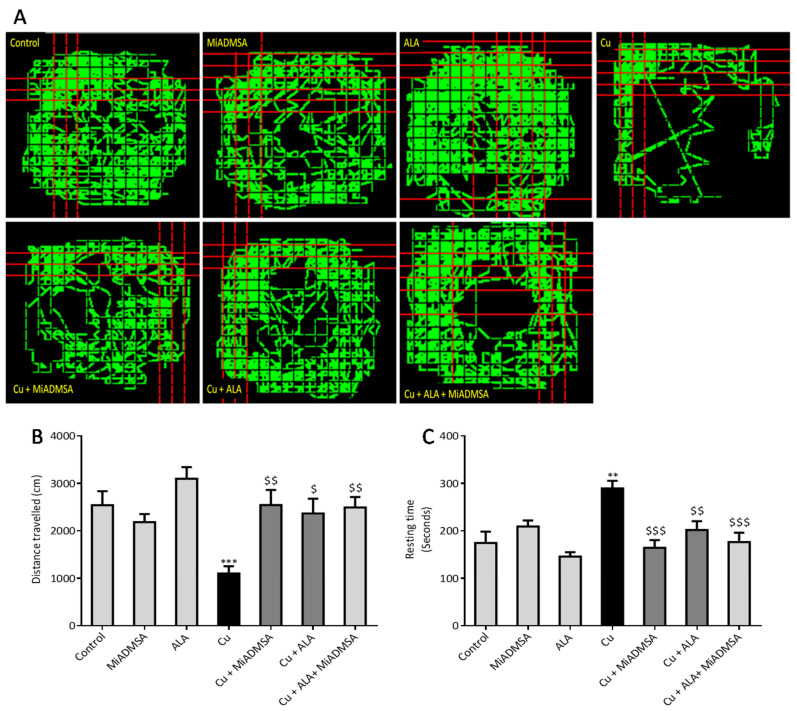
The MiADMSA and ALA combination therapy attenuated spontaneous locomotor activities. (**A**) Tracking graphs; (**B**) Distance travelled; (**C**) Resting time. All the values are expressed as mean ± S.E.M. (*n* = 6), *** *p* < 0.001 vs. control and $ *p* < 0.05, $$ *p* < 0.01, $$$ *p* < 0.001 vs. Cu. **: It indicates the comparison between the groups which we have mentioned in the figure legends.

**Figure 9 toxics-10-00718-f009:**
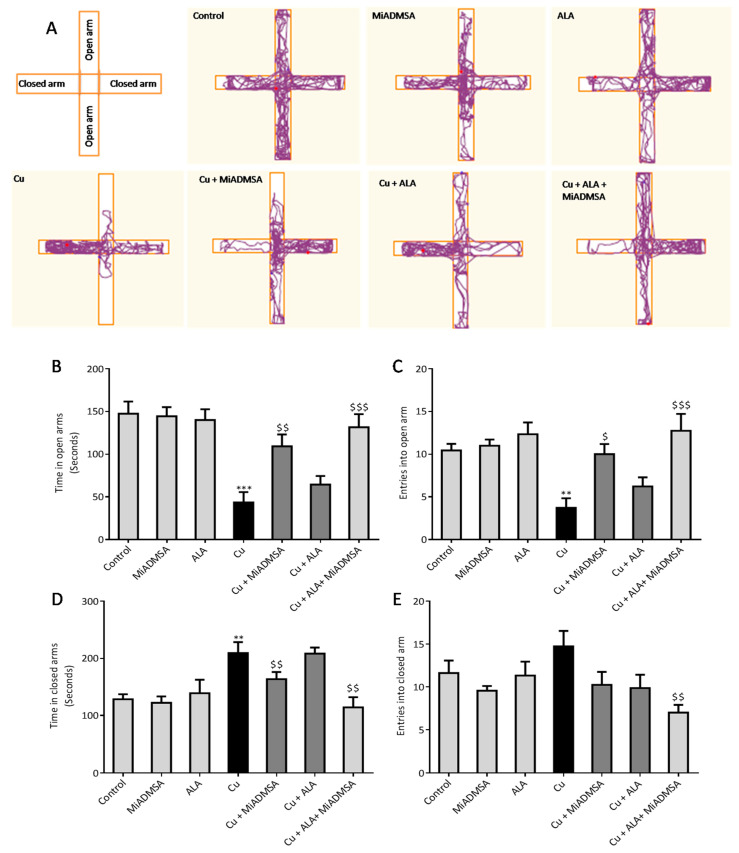
The combination therapy with MiADMSA and ALA improved the elevated plus-maze performance. (**A**) Tracking graphs; (**B**) Time in open arms; (**C**) Entries into open arms; (**D**) Time in closed arms; (**E**) Entries into closed arms. All the values are expressed as mean ± S.E.M. (*n* = 6) * *p* < 0.05, *** *p* < 0.001 vs. control and $ *p* < 0.05, $$ *p* < 0.01 vs. Cu. $$$: It indicates the comparison between the groups which we have mentioned in the figure legends. **: It indicates the comparison between the groups which we have mentioned in the figure legends.

**Figure 10 toxics-10-00718-f010:**
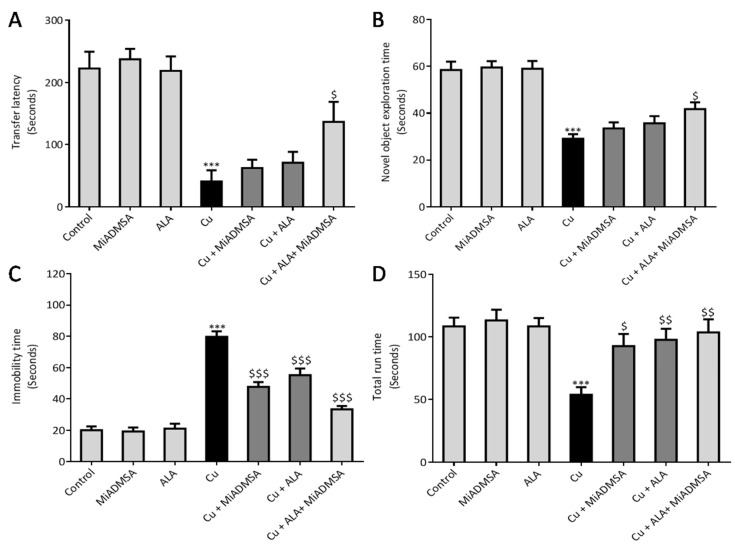
Protective effect of combination therapy of MiADMSA and ALA on memory and neurobehavioral functions. (**A**) Transfer latency (passive avoidance test); (**B**) NORT test; (**C**) Forced swim test; (**D**) Total run time in rotarod. All the values are expressed as mean ± S.E.M. (*n* = 6), *** *p* < 0.001 vs. control and $ *p* < 0.05, $$ *p* < 0.01, $$$ *p* < 0.001 vs. Cu.

**Figure 11 toxics-10-00718-f011:**
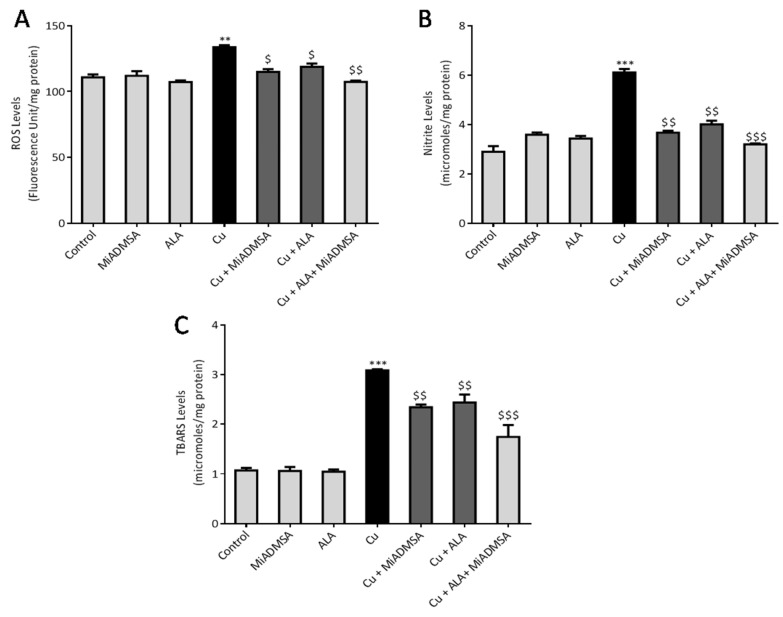
The MiADMSA and ALA combination treatment mitigates the augmented brain ROS, nitrite, and TBARS levels. (**A**) Brain ROS level; (**B**) Brain nitrite level; (**C**) Brain TBARS level. All the values are expressed as mean ± S.E.M. (*n* = 6) ** *p* <0.01, *** *p* < 0.001 vs. control and $ *p* < 0.05, $$ *p* < 0.01, $$$ *p* < 0.001 vs. Cu.

**Figure 12 toxics-10-00718-f012:**
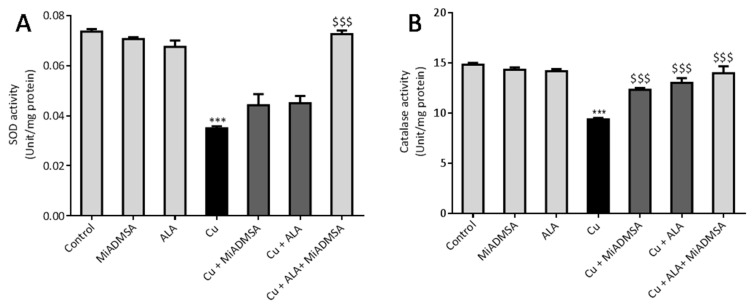
The MiADMSA and ALA combination therapy restored the altered brain SOD and catalase levels. (**A**) Brain SOD level; (**B**) Brain catalase level. All the values are expressed as mean ± S.E.M. (*n* = 6), *** *p* < 0.001 vs. control and $$$ *p* < 0.001 vs. Cu.

**Figure 13 toxics-10-00718-f013:**
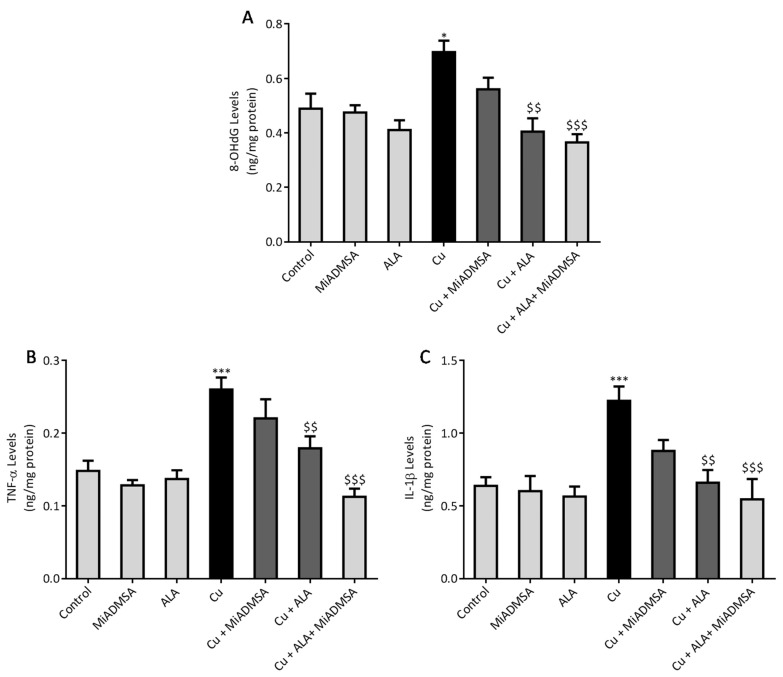
The combination therapy of MiADMSA and ALA diminished the brain 8-OHdG, TNF-α, and IL-1β levels. (**A**) Brain 8-OHdG level;(**B**) Brain TNF-α level; (**C**) Brain IL-1β level. All the values are expressed as mean ± S.E.M. (*n* = 6), *** *p* < 0.001 vs. control and $$ *p* < 0.001, $$$ *p* < 0.001 vs. Cu. *: It indicates the comparison between the groups which we have mentioned in the figure legends.

## Data Availability

Not applicable.

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
