# Peer review of "Alpha Lipoic Acid and Monoisoamyl-DMSA Combined Treatment Ameliorates Copper-Induced Neurobehavioral Deficits, Oxidative Stress, and Inflammation"

_toxics, 2022, doi:10.3390/toxics10120718_

Round 1
Reviewer 1 Report
The manuscript by Patwa, Thakur and Flora describes the role of two chemical compounds chelating Cu ions that might be used as the detoxifying agents after enormous exposure to Cu. I have no doubts that all experimental set-up is designed properly and the results obtained are very important from the redox-derived neurodegeneration point of view. I think that the paper should be published in Toxics. However, I have some questions and comments to the authors.
1. What is the level of Cu in blood of Cu-treated animals? Have you ever tested it? It might be the effect of Cu action from the extracellular matrix rather than from the cell interior. If the authors suggest Cu accumulation in neurons and other cells of brain tissue, it would be valuable to find accumulation sites of Cu using transmission electron microscopy. Could you comment on possible sites of Cu accumulation in discussion? Neurons, glial cells, endothelial cells etc.? The animals were treated with Cu for 16 weeks followed by ALA and MiADMSA administration. I understood that the authors expected the chelation activity of the chemicals in relation to Cu ions present in cells and tissues. But, if the Cu ions are in the bound form inside the cells (accumulated), how do the chelators act on Cu?
3. Could you add information about chelation potential of both compounds in relation to other 1+/2+ or 2+/3+ ions (e.g. Fe+) that are also physiologically important for cell functioning? Is there any interaction with other ions?
4. There is no Control group of animals treated with ALA and MiADMSA. Why?
5. Add to the discussion the potential role of chelating agents in neurodegeneration protection. Should we apply the chelators during our life or when any symptoms of neurogeneration appear?
6. Add to the introduction the potential group of people/animals with high risk of exposure to copper that could be treated with the chelators when exposed acutely or chronically to high concentrations of Cu ions. Copper miners?
Author Response
Reviewer 1
The manuscript by Patwa, Thakur, and Flora describes the role of two chemical compounds chelating Cu ions that might be used as detoxifying agents after enormous exposure to Cu. I have no doubts that all experimental set-up is designed properly and the results obtained are very important from the redox-derived neurodegeneration point of view. I think that the paper should be published in Toxics. However, I have some questions and comments for the authors.
Reply: We thank the reviewer for his/her positive and critical evaluation of the manuscript. Below is our response to the specific comments made by the learned reviewer.
Comment# 1. What is the level of Cu in the blood of Cu-treated animals? Have you ever tested it? It might be the effect of Cu action from the extracellular matrix rather than from the cell interior. If the authors suggest Cu accumulation in neurons and other cells of brain tissue, it would be valuable to find accumulation sites of Cu using transmission electron microscopy. Could you comment on possible sites of Cu accumulation in the discussion? Neurons, glial cells, endothelial cells, etc.? The animals were treated with Cu for 16 weeks followed by ALA and MiADMSA administration. I understood that the authors expected the chelation activity of the chemicals about Cu ions present in cells and tissues. But, if the Cu ions are in the bound form inside the cells (accumulated), how do the chelators act on Cu?
Reply: We regret that blood Cu level was not determined in the current primarily for the reason that numerous published reports are available which clearly suggest the significant Cu accumulation in the brain. MiADMSA for being lipophilic in nature is known to cross the cell membrane and deplete Cu form extracellular matrix through competitive binding. Cu ions are known to have strong binding toward the thiol molecules.
Comment# 2. Could you add information about the chelation potential of both compounds about other 1+/2+ or 2+/3+ ions (e.g. Fe+) that are also physiologically important for cell functioning? Is there any interaction with other ions?
Reply: MiADMSA has thiol molecules that are responsible for chelation of Cu. Our earlier published study on the effect of MiADMSA on other essential elements suggests that MiADMSA did not alter the essential elements like iron, zinc, magnesium, and calcium in blood however Cu levels get decreased. Thus, it could be of huge advantage over other chelating agents as it does not interfere with other bio-metals. The information has now been included in the revised manuscript [Kindly see the highlighted portion line no.68-72]
Comment# 3. There is no Control group of animals treated with ALA and MiADMSA. Why?
Reply: We understood that these groups need to be there but the learned reviewer will appreciate that number of our previous reports have investigated the individual effects of MiADMSA and LA. As we have clearified in the text that MiADMSA has been approved by Drug Controller General of India, thus all the data for MiADMSA per se, using variable does following acute and chronic exposure have been generated. Thus in order to refine the unnecessary use of animals was avoided. [1].
1) Flora, S.J.S., Mehta, A., Gautam, P., Jatav, P., Pathak, U. (2007) Essential metal status, prooxidant/antioxidant effects of MiADMSA in male rats: age-related effects. Biol. Trace Elem. Res.120, 235-247.
2) Mehta, A. et al Hematological, hepatic and renal alterations after repeated oral or intraperitoneal administration of monoisoamyl DMSA I. Changes in Male Rats’ J. Appl. Toxicol. 22, 359-369, 2002
3) Flora S.J.S., Mehta, A. Hematological, hepatic and renal alterations after repeated oral or intraperitoneal administration of monoisoamyl DMSA II. Changes in Female Rats’ J. Appl. Toxicol. 23, 97-102, 2003
4) Mehta, A., Pant S.C., Flora, S.J.S. Monoisoamyl dimercaptosuccinic acid induced changes in pregnant female rats during late gestation and lactation. Reproductive Toxicology, 21, 94-103, 2006
5) Pande M, and Flora S J S, Lead induced oxidative damage and its response to combined administration of -Lipoic acid and succimers in rats, Toxicology, 177, 187-196, 2002\
6) Bhatt K, Flora S.J.S. Oral co-administration of - lipoic acid, quercetin and captopril prevents gallium arsenide toxicity in rats. Environmental Toxicology and Pharmacology, 28, 140-146, 2009
Comment# 4. Add to the discussion the potential role of chelating agents in neurodegeneration protection. Should we apply chelators during our life or when any symptoms of neurodegeneration appear?
Reply: We have incorporated more about the protective role of chelating agents’ neurodegeneration in the discussion. [Kindly see the highlighted sentences line no. 584-586]
Comment# 5. Add to the Introduction the potential group of people/animals with a high risk of exposure to copper that could be treated with the chelators when exposed acutely or chronically to high concentrations of Cu ions. Copper miners?
Reply: We added the suggested content in the introduction section [Kindly see the highlighted sentences line no.38-42].
Reviewer 2 Report
Comments "This is an impressive piece of work when it comes to the amount of experiments and analyses performed. However, the study suffers from its single-dose design. A dose response approach of ALA and MiADMSA, respectively, would have contributed to more easily interpreted data. As presented within the manuscript, it is not possible to draw any conclusion whether the increased efficacy of the combination is due to simple additive effects of the two compounds or due to a synergistic effect of the compounds. Furthermore, findings that 75 mg/kg ALA plus 75 mg/kg MiADMSA, respectively, in general caused more pronounced responses than each of them alone at a dose of 75mg/kg are rather expected findings. As the present experiments did not include 150mg/kg of ALA or MiADMSA, one cannot conclude "that combined administration of MiADMSA and ALA might be a better treatment strategy than monotherapy". EPM, PMS, NBT and 8-OHdG lacking in the list of abbreviations (line 567). There are typographical/grammatic errors, particularly in the Discussion/Conclusion section, for example at line 482-484, 536-537, 539-540 and 557-558. ROS/RNS stands for Reactive Oxygen Species/Reactive Nitrogen Species (531). Legend figure 3 lacking "(A)" and "(B)" (line 326)"
Author Response
Reviewer # 2
Comments "This is an impressive piece of work when it comes to the number of experiments and analyses performed. However, the study suffers from its single-dose design.
Reply: Thank you very much for the encouraging words and critical suggestions to improve the quality and clarity of the present manuscript.
Comment # 1 A dose-response approach of ALA and MiADMSA, respectively, would have contributed to more easily interpreted data. As presented within the manuscript, it is not possible to draw any conclusion whether the increased efficacy of the combination is due to the simple additive effects of the two compounds or due to a synergistic effect of the compounds. Furthermore, findings that 75 mg/kg ALA plus 75 mg/kg MiADMSA, respectively, in general, caused more pronounced responses than each of them alone at a dose of 75mg/kg are rather expected findings. As the present experiments did not include 150mg/kg of ALA or MiADMSA, one cannot conclude "that combined administration of MiADMSA and ALA might be a better treatment strategy than monotherapy".
Reply: We agreed with the reviewer’s suggestion that a dose dependent study using variable doses of MiADMSA and/or LA should have been used. However, study protocol especially the doses were chosen considering previous reports from literature and our own group. ALA has shown a dose-dependent protective effect in terms of amelioration of oxidative stress parameters. Though, our previous finding suggests that MiADMSA at the dose of 150 mg/kg led to some liver related anomalies. Thus, based on our previous experience we opted for a single dose of MiADMSA along with ALA as it has anti-oxidant, metal chelation, and anti-inflammatory activity. However, in our future studies, we would try to address the dose-dependent effects of combination therapy of MiADMSA and ALA [1].
1) Pande, M., Flora, S. J.S. (2002) Lead induced oxidative damage and its response to combined administration of alpha-lipoic acid and succimers in rats. Toxicol. 177, 187-196.
Comment # 2, PMS, NBT, and 8-OHdG lacking in the list of abbreviations (line 567). There are typographical/grammatical errors, particularly in the Discussion/Conclusion section, for example in lines 482- 484, 536-537, 539-540, and 557-558. ROS/RNS stands for Reactive Oxygen Species/Reactive Nitrogen Species (531). Legend figure 3 lacking "(A)" and "(B)" (line 326)"
Reply: We have thoroughly reviewed the entire manuscript for the English Language/grammatical errors, and appropriate editing/corrections have been made in the revised manuscript. [Kindly the text highlighted with yellow color in the revised manuscript].
Reviewer 3 Report
In the present study, the authors investigated the combined effects of monoisoamyl 2, 3-dimercaptosuccinic acid (MiADMSA), a new chelator, and alpha lipoic acid (ALA), an antioxidant that is made naturally in the body and is also found in foods, against copper- (Cu)-induced oxidative stress in rats.
This study has an interesting topic, unfortunately, this manuscript has a lot of serious inaccuracies and methodological mistakes and needs very very intensive improvements and corrections before publishing may be possible.
Special points:
Please do all citations in the text according to “Toxics”.
Please do your List of references at the end of the manuscript according to “Toxics”.
Keywords: please add also to keywords: rats.
Introduction
Lines 31-49: please add multiple references at the end of each of these sentences.
Lines 50-56: please add multiple references at the end of each of these sentences.
Lines 65-71: please describe all these studies exactly.
Please describe very exactly in your Introduction section all already published similar studies from your lab and from the literature.
Materials and Methods
Please add to each section, each method according to which author or scientific group you did this method. Please also add appropriate references for each method. Please add these references at the beginning of each section.
Please add the exact number of the animals used for each method/section of each experimental group.
Animals
Please add the total number of the animals used in your study.
Please add the organisation name, the exact date and protocol number for the permission of all your experiments.
Please say “g “and not “gm” in the whole manuscript.
Experimental Design
Please put the section “Experimental Design” after the “Animals” section.
Please describe all these studies very exactly in your Introduction section: 21, 24, 17.
Assessment of neurobehavioral functions
What about the adaptation of the animals to the experimental room before training or testing?
Once again, please add to each section, each method according to which author or scientific group you did this method. Please also add appropriate references for each method. Please add these references at the beginning of each section.
Open field test
Which monitoring system did you use exactly for this method?
What about the disinfection of the open field box after each rat?
Please add also the anxiety results from this test, not only the motoric results.
Rotarod test
Please include in your results also the training results as a Diagram.
Elevated plus maze
Please add the exact product information to this apparatus.
Please include in your results also the training results as a Diagram. Please use both open field test and elevated plus maze test to analyse the anxiety of the animals.
Forced swim test
Please describe this section very exactly.
Please say, which tank exactly you used, how high and how wide was it for this test? From which company was this tank? To say: “a cylindrical tank containing half-filled water“ is not enough, please say this in cm.
What about the temperature of this water?
To analyse the immobility time with a stopwatch is not possible, this method is not acceptable. You need to analyse this only with special video tracking system.
Please add also the results for nother parameters of this test as a diagram.
What about the care of the animals after swimming in the water? Did you change the water, how often?
How long were animals in the water, how long lasts this test?
Nobel object recognition test
Please add the exact product information to this apparatus.
Which tracing system did you use to analyse these results?
Passive avoidance test
Please add the exact product information of this apparatus.
Estimation of brain AChE activity
What about the anaesthesia of the animals before brain perfusion?
Statistical analysis
Please describe this section more exactly.
Results
Figure 2: please add a Legend with the description to this Figure.
Figure 3: please add a Legend with the description to this Figure.
Figure 4: please add a Legend with the description to this Figure.
Figure 5: please add a Legend with the description to this Figure.
Figure 6: please add a Legend with the description to this Figure.
Discussion
Please describe all results of the behavioural test separately und please discuss all these results step by step and separately.
Lines 459-466: please add multiple references at the end of each of these sentences.
Lines 467-471: please describe all these studies exactly.
Conclusions
Please add the Future perspectives section to your manuscript.
Author Response
Reviewer # 3
Comments and Suggestions for Authors
In the present study, the authors investigated the combined effects of monoisoamyl 2, 3-dimercaptosuccinic acid (MiADMSA), a new chelator, and alpha lipoic acid (ALA), an antioxidant that is made naturally in the body and is also found in foods, against copper- (Cu)-induced oxidative stress in rats. This study has an interesting topic, unfortunately, this manuscript has a lot of serious inaccuracies and methodological mistakes and needs very intensive improvements and corrections before publishing may be possible.
Reply: Thank you very much for the critical comments which will be helpful in improving our manuscript. We have carefully gone through each comments and have incorporated them in the revised version of our manuscript particularly providing a more rigorous methodology. We hope that the revised manuscript will now be acceptable.
Special points:
Please do all citations in the text according to “Toxics”.
Please do your List of references at the end of the manuscript according to “Toxics”.
Keywords: please add also to keywords: rats.
Reply: Suggestion incorporated
Introduction
Lines 31-49: please add multiple references at the end of each of these sentences.
Reply: Suggestion incorporated
Lines 50-56: please add multiple references at the end of each of these sentences.
Reply: Suggestion incorporated.
Lines 65-71: Please describe all these studies exactly. Please describe very exactly in your Introduction section all already published similar studies from your lab and the literature.
Reply: Suggestion incorporated in the revised manuscript.
Materials and Methods
Please add to each section, each method according to which author or scientific group you did this method. Please also add appropriate references for each method.
Reply: Suggestions incorporated in the revised manuscript.
Please add these references at the beginning of each section.
Reply: Suggestion incorporated.
Please add the exact number of animals used for each method/section of each experimental group.
Reply: Suggestions incorporated.
Animals
Please add the total number of animals used in your study.
Reply: Revised accordingly.
Please add the organization name, the exact date, and the protocol number for the permission of all your experiments.
Reply: Revised accordingly.
Please say “g “and not “gm” in the whole manuscript.
Reply: Suggestion incorporated.
Please describe all these studies very exactly in your Introduction section: 21, 24, 17.
Reply: Suggestion incorporated.
Assessment of neurobehavioral functions
What about the adaptation of the animals to the experimental room before training or testing? Once again, please add to each section, each method according to which author or scientific group you did this method. Please also add appropriate references for each method.
Please add these references at the beginning of each section.
Reply: Revised accordingly.
Open field test
Which monitoring system did you use exactly for this method? What about the disinfection of the open field box after each rat?
Please add also the anxiety results from this test, not only the motoric results.
Reply: We have revised the method accordingly.
Elevated plus maze Please add the exact product information to this apparatus.
Reply: Revised accordingly.
Forced swim test
Please describe this section very exactly. Please say, which tank exactly you used, how high and how wide was it for this test? From which company was this tank? To say: “a cylindrical tank containing half-filled water“ is not enough, please say this in cm. What about the temperature of this water? To analyze the immobility time with a stopwatch is not possible, this method is not acceptable. You need to analyze this only with a special video tracking system. Please add also the results for other parameters of this test as a diagram. What about the care of the animals after swimming in the water? Did you change the water, and how often? How long were animals in the water, and how long lasts this test?
Reply: Now we have described the forced swim test as suggested.
Nobel object recognition test
Please add the exact product information to this apparatus. Which tracing system did you use to analyze these results?
Reply: Suggestion incorporated.
Passive avoidance test
Please add the exact product information of this apparatus.
Reply: Information has been included in the revised manuscript.
Estimation of brain AChE activity What about the anesthesia of the animals before brain perfusion?
Reply: We have not perfused the rat’s brain.
Statistical analysis
Please describe this section more exactly.
Reply: Now we have described more precisely the statistical method that we have used in the current investigation.
Results
Figure 2: please add a Legend with the description to this Figure.
Figure 3: please add a Legend with the description to this Figure.
Figure 4: please add a Legend with the description to this Figure.
Figure 5: please add a Legend with the description to this Figure.
Figure 6: please add a Legend with the description to this Figure.
Reply: Now we have added more information in the figure legend [kindly see the figure no 2 to 6].
Discussion Please describe all results of the behavioral test separately and please discuss all these results step by step and separately.
Lines 459-466: please add multiple references at the end of each of these sentences.
Reply: Suggestion incorporated in the revised manuscript.
Lines 467-471: Please describe all these studies exactly.
Reply: Now we have described the suggested studies in the revised manuscript
Conclusions Please add the Future perspectives section to your manuscript.
Reply: We have added the future perspectives in the revised manuscript [kindly see the conclusion paragraph]
*****************
Round 2
Reviewer 2 Report
I have gone through the revised manuscript and concluded that the original Comment #1 should be addressed in the Discussion section as a limitation in line with what is said in the corresponding authors' Reply.
Author Response
Comment - I have gone through the revised manuscript and concluded that the original Comment #1 should be addressed in the Discussion section as a limitation in line with what is said in the corresponding authors' Reply.
Reply- As suggested by the reviewer # 2, we have now included a paragraph (kindly see the last paragraph of Discussion) in the Discussion suggesting the limitation of this study and that future studies will be conducted using variable dose of LA and MiADMSA.